# Chitosan: An Autocidal Molecule of Plant Pathogenic Fungus

**DOI:** 10.3390/life12111908

**Published:** 2022-11-16

**Authors:** Debanjana Debnath, Ipsita Samal, Chinmayee Mohapatra, Snehasish Routray, Mahipal Singh Kesawat, Rini Labanya

**Affiliations:** 1Department of Plant Pathology, Faculty of Agriculture, Sri Sri University, Cuttack 754006, Odisha, India; 2Department of Entomology, Faculty of Agriculture, Sri Sri University, Cuttack 754006, Odisha, India; 3Department of Genetics and Plant Breeding, Faculty of Agriculture, Sri Sri University, Cuttack 754006, Odisha, India; 4Department of Soil Science & Agricultural Chemistry, Faculty of Agriculture, Sri Sri University, Cuttack 754006, Odisha, India

**Keywords:** chitosan, pathogen, sustainable, plant protection

## Abstract

The rise in the world’s food demand with the increasing population threatens the existence of civilization with two equally valuable concerns: increase in global food production and sustainability in the ecosystem. Furthermore, biotic and abiotic stresses are adversely affecting agricultural production. Among them, losses caused by insect pests and pathogens have been shown to be more destructive to agricultural production. However, for winning the battle against the abundance of insect pests and pathogens and their nature of resistance development, the team of researchers is searching for an alternative way to minimize losses caused by them. Chitosan, a natural biopolymer, coupled with a proper application method and effective dose could be an integral part of sustainable alternatives in the safer agricultural sector. In this review, we have integrated the insight knowledge of chitin-chitosan interaction, successful and efficient use of chitosan, recommended and practical methods of use with well-defined doses, and last but not least the dual but contrast mode of action of the chitosan in hosts and as well as in pathogens.

## 1. Introduction

Global climatic changes are posing a threat to the security of the world’s food supply by adversely affecting plant growth, development, and yield in multiple directions, such as by causing abiotic stresses to plants and as well as encouraging and strengthening the biotic populations by increasing their resistance against conventional chemical management procedures. In this situation, the researchers are trying to identify some natural compounds or their derivatives which can be effectively established themselves as an ideal one to replace the chemicals against which the pathogens are growing resistance gradually. Chitosan, a chemically and physically diverse compound and long-chain polymer of *N*-acetyl-glucosamine and d-glucosamine derivative of chitin (second most prevalent polysaccharide after cellulose), was first discovered in 1859 by Rouget [1]. Due to its potential for usage in antiviral, antifungal, and antibacterial products, chitosan and its derivatives have gained attention in recent years. Other than chitosan, some other chitin-related compounds and chitin derivatives have also been identified as possible plant protection agents [2]. Chitosan, an aminoglucopyranan made up of N-acetyl-D-glucosamine (GlcNAc) and glucosamine (GlcN) residues, is currently being deemed essential due to its appealing features and biological activities [3,4]. An amino group and two hydroxyl groups make up the three reactive functional groups that make up chitosan. 

According to Amine et al. 2021 chitin and its derivative chitosan are effective at boosting plant defence against pathogens in monocotyledons and dicotyledons [5]. Cell wall lignification, cytoplasmic acidification, membrane depolarization, changes in ion flux and protein phosphorylation, phytoalexin biosynthesis, production of reactive oxygen species, jasmonic acid biosynthesis, and activation of chitinase and glucanase are all targets of this increased plant defense [6].In addition to this, many dicot species have been observed to produce callose, proteinase inhibitors, and phytoalexin in response to chitosan that also have significant roles in plant defense [7]. In the current situation, the rising incidences of resistance, residue, and resurgence (3Rs) have facilitated the increased use of naturalytes in disease and pest control [8]. Consequently, chitosan, a byproduct of fungal and insect chitin, may be used as an autocidal agent to kill invasive disease causing pathogen. 

## 2. Structure and Formation of Chitin

Chitin, a β (1,4)-linked homopolymer of N-acetylglucosamine, is a simple polysaccharide that is an important component of fungal cell wall [9]. It is an amino sugar biopolymer that forms elaborate structures such as insect cuticles and peritrophic membranes when combined with a range of proteins. This polymer is mostly used as a structural component, and it is similar to cellulose and collagen in plants and vertebrates, respectively [10]. 75% of the overall weight of shellfish including crab and shrimp generally discarded as waste among which 20-58% is made up of chitin [11].

Broadly, chitin is defined as a β-(1–4) linked linear cationic heteropolymer consisting of 2-acetamide-2-deoxy-D-glucopyranose (N–acetyl–D–glucosamine, GlcNAc) and randomly distributed units of 2-amino-2-deoxy-D-glucopyranose (D–glucosamine, GlcN) [12]. Due to the presence of the acetoamide groups in the *trans* position, chitin exhibits two important properties: a high degree of crystallinity and a lack of solubility in water and organic solvents [13].

X-ray diffraction analysis revealed that chitin generally occurs in three different crystalline forms, termed α-, β-, and γ-chitin [10], which mainly differ from each other in the degree of hydration, size of the unit cell, and the number of chitin chains per unit cell [14]. All the chains exhibit an anti-parallel orientation in the α form, whereas in the β form the chains are arranged in a parallel manner and in the γ form sets of two parallel strands alternate with single anti-parallel strands. All three crystalline forms are primarily found in the chitinous structures of insects. The α form is most prevalent in chitinous cuticles, whereas the β and γ forms are frequently found in cocoons [15,16].

Due to the properties such as biodegradability, biocompatibility, non-toxicity, physiological inertness, and gel-forming properties, chitin has been found to have countless applications in different industries, e.g., food, cosmetic, pharmaceutical, manufacturing of synthetic materials, agriculture, and even electronics for the production of biosensors [17,18].

In the first step, glycogen is converted by glycogen phosphorylase to glucose-1-P, which is either fed into glycolysis or used for trehalose synthesis [19]. Additionally, the enzyme trehalase can mobilise trehalose by hydrolyzing it to glucose. This is followed by the enzymes hexokinase, phosphoglucomutase, and glucose-6-P isomerase converting glucose to fructose-6-P. Finally, from fructose-6-P the chitin biosynthetic pathway branches off, with the first enzyme catalyzing this branch being glutamine-fructose-6-phosphate amidotransferase [10]. The reaction catalyzed by GFAT converts fructose-6-P into glucosamine-6-phosphate by transferring the ammonia from the co-substrate l-glutamine and isomerizing the resulting fructosimine-6-phosphate. Next, an acetyl group from coenzyme A is added by glucosamine-6-P acetyltransferase to obtain N-acetylglucosamine (GlcNAc)-6-P [20] (Figure 1), whose phosphate is then transferred from the C-6 to the C-1 position catalyzed by phosphoacetylglucosamine mutase. The resulting GlcNAc-1-P, finally, is uridinylated by UDP-GlcNAc pyrophosphorylase yielding UDP-GlcNAc which serves as a substrate for the chitin synthase, transferring the sugar moiety of UDP-GlcNAc to the growing chitin chain. Chitin is degraded by chitinases and N-acetylglucosaminidases yielding GlcNAc which can be reused for chitin biosynthesis [21].

### Deacetylation and Hydrolysis of Chitin/Chitosan

Unfortunately, despite having so many advantages, there is very limited use of chitin due to resistance to different physical and chemical agents because of its highly ordered crystalline structure and its lack of solubility in water or any organic solvents. Chitosan, an N-deacetylated derivative of chitin that is soluble in aqueous solutions of both organic and inorganic acids, is frequently employed to get around this restriction [22]. This cationic polymer resembles chitin and also consists of β-(1–4) linked N–acetyl–D–glucosamine and the D–glucosamine residues [23]. Chitosan, which is produced industrially by hydrolyzing the amino acetyl groups of chitins, is less frequent in nature than chitin. It is also naturally present in the cell walls of filamentous fungi primarily belonging to the *Zygomycetes* class [24]. Carboxymethyl-chitin (CM-chitin), as a water-soluble anionic polymer, is the second most studied derivative of chitin after chitosan. Most of the chitin and chitosan biological properties are directly related to their physicochemical characteristics. These characteristics include degree of deacetylation, molecular mass, and the amount of moisture content [25].

Enzymatic degradation of chitin can be achieved by two different paths:Chitin can be degraded by first being solubilized by deacetylation (Figure 2). This process is carried out by chitin deacetylases, and the derived substrate (chitosan) is hydrolysed by chitosanases [18,26].The chitinolytic process requires direct hydrolysis of the beta-1,4 glycosidic bonds between the GlcNAc units by chitinases. Chitinases are produced by higher plants, which use the enzymes to defend themselves against pathogenic attacks by degrading chitin in the cell walls of fungi and bacteria [27]. Plant chitinases have molecular weights ranging from 25 to 40 kD and can be acidic or basic. Endochitinases and exochitinases are the two types of chitinases [28]. Chitinase genes from biocontrol fungi such as *Trichoderma* have significantly higher antifungal activity than comparable plant genes. These fungal genes encode for chitinolytic enzymes, which have higher antifungal activity similar to chemical fungicides [29].

Despite having many significant similarities in the molecular structures of chitin and chitosan, the physicochemical characteristics of both the biopolymers and the reactions are often surprisingly distinct [18]. Both polymers have the same reactive hydroxyl and amino groups in different molecular ratios, but the lower crystallinity of chitosan makes it more accessible to reagents [12]. Perhaps the most important and crucial difference between chitin and chitosan in terms of their applications is based on their degree of deacetylation and their solubility. Unlike chitin, chitosan has a below pKa (pH~6.5) in most acidic aqueous solutions such as acetic acid, formic acid, lactic acid, citric acid, and other solvents such as dimethylsulfoxide and p-toluenesulfonic acid [30].

## 3. Successful Use of Chitin/Chitosan against Plant Pathogen with Special Reference of Pathogenic Fungus

With changing times and increasing knowledge, producers have begun to identify alternatives to toxic chemicals that are harmful not only to consumers but also to the environment and ecosystems. In these circumstances, use of chitosan against plant pathogens is receiving popularity and wider acceptance due to its eco-friendly nature and abundance of its source [7]. The irony is that chitin and chitosan are obtained from the fungus or the insect and this molecule has unique characteristics to cause damage to multiple plant pathogens, even including fungi itself singly or in combination which make them an autocidal component for fungi (Table 1). A combination of allicin (5% allicin ME 100–time dilution liquid) + chitosan (100–time dilution liquid) showed 85.97% control effect against powdery mildew which was significantly (*p* < 0.01) higher than allicin (76.70% of) and chitosan (70.93%) alone [31]. Some recent research suggesting the use of chitin/chitosan has been mentioned below.

## 4. Application of Chitosan

After discovering chitosan’s antipathogenic activity, especially its antifungal activity, the second major concern of researchers was how to use chitosan effectively. The appropriate way of application is directly proportional to the rate of success and higher efficiency. On the other hand, application methods should be easy and cheap in nature.

### 4.1. Seed Treatment

Use of chitosan in various dose dependent manners has been proved to be very effective against different pathogens (Table 2). Chitosan not only creates a barrier between pathogens and healthy embryos, but it also helps retain moisture which increases germination rate and also affects plant vitality. effectively combats pathogens and also induces defense in seedlings grown from treated seeds [41]. 

### 4.2. Chitosan Used for Soil Amendment

The use of chitosan as a soil conditioner or soil treatment has enormous direct or indirect effects on plant pathogens. There are many experiments performed by researchers who noted different activities of chitosan against different pathogens.Corsee et al. (2015) demonstrated a new concept of chitosan-induced plant defense mechanism, in which kiwi plant immunity was enhanced by applying chitosan to the growth medium by acting as a trigger to increase the activity of guaiacol peroxidase. (G-POD), ascorbate peroxidase (APX), phenylalanine ammonia lyase (PAL) and polyphenol oxidase (PPO) that regulate plant defense [44]. Moreover, adding chitosan to soil promotes the growth and abundance of beneficial microorganisms such as Pseudomonas fluorescens, actinomycetes, mycorrhizal fungi and rhizobia [45,46]. Both chitin and chitosan are taken up by soil at different rates, and chitosan acts as chitin when parasitic on tomato fields [47] and neither chitosan nor chitin showed phytotoxicity to the host.

### 4.3. Chitosan Used as Foliar Spray

Foliar sprays are known for their ease of use and direct contact with infected pathogens and the symptoms they cause. They also create a protective barrier between pathogen and host, kill contact-associated pathogens, impede fungal sporulation, and most importantly help the host to induce defense mechanisms [40,48]. Some of the examples has been cited below in Table 3.

### 4.4. Chitosan Used as Post Harvest Fruit Treatment

Chitosan shows a dual mode of action in post-harvest disease control where it reduces the growth of decay-causing fungi and foodborne pathogens and induces resistance responses in the host tissues (Table 4). Chitosan coating forms a semipermeable film on the surface of fruit and vegetables, thereby delaying the rate of respiration, decreasing weight loss, maintaining the overall quality, and prolonging the shelf life.

### 4.5. Effect of Chitosan in Plant Disease Control

In combating infectious diseases, chitosan exhibits different mechanisms of action against various pathogens (Figure 3). Some of them are directly related to inhibiting pathogen growth and multiplication, while others are involved in activating or enhancing plant defenses to combat pathogens and achieve sustainable yields. Thus, it is possible to produce visible positive changes in the host by improving yield-related parameters and to enhance the plant’s own defenses so that the host can resist attack by plant pathogens, is the dual nature of chitosan’s action. On the other hand, chitosan adversely affects or causes negative changes in pathogen growth and fertility.

Indirect treatment of disease by activating defenses and improving plant health

Different methods of chitosan treatment enhance plant defenses by producing various defense enzymes, proteins and phytorexins. Sidia et al. In 2018, mentioned broader ideas regarding the defense-related activities of seed treatments with chitosan nanoparticles (CNP) [58]. CNP treatment expressed high levels of pathogenesis-related proteins PR1 and PR5. Seedlings treated with chitosan nanoparticles increased maximal phenylalanine ammonia lyase (PAL), polyphenol oxidase (PPO) and peroxidase (POX) activities by 1.08-fold, 1.10-fold and 1.10-fold, respectively. with normal chitosan treatment. It induced both systemic and permanent tolerance and showed significant protection against downy mildew. Treatment with chitosan also increased plant vigor, recording 89% seed germination. All this leads to a lower downy mildew incidence in the treated plants. H. 18.1% and 19.6% for CNP and chitosan treatment, respectively. Zen et al. (2010) also noted increases in peroxidase (POD) and superoxide dismutase (SOD) activities, as well as increases in glutathione (GSH) and hydrogen peroxide (H2O2) levels following post-harvest treatment of fruit with the application of chitosan on Navel Orange (Citrus sinensis L. Osbeck)[59]. Peroxidase and poly-phenol oxidase are responsible for eliciting plant defence. Peroxidase activity and peroxidase gene expression both were increased by chitosan treatment in many folds comparable to the control [56,60]. Chitosan significantly increases polyphenol oxidase activity by catalysing the phenolic substances responsible for lignin synthesis, that gives strength to the host cell wall and promotes prevention of pathogen entry [61,62]. Catalase enzyme responsible for degradation of H_2_O into H_2_O_2_ and O_2_ increased by the chitosan treatment in peach indicates that it plays a distinct responsibility in increasing defense, controlling aging and senescence [56]. In contrast, chitosan treatment can cause some visible impact on plants. Seed germination rate, i.e., 94.45% was increased by application of 5% chitosan by creating a semi-permeable membrane on the seed surface that helps maintain seed moisture and improve germination [63]. A combination of seed treatment and foliar application also helps increase plant height and yield [38]. With more vigorous disposition and improved yield parameters, these seeds help plants maintain yields and fend off attack by pathogens.

2.Direct method of pathogen control.

a. Chitosan mediated detrimental changes in plant pathogen

Chitin of fungal cell wall hydrolyses by chitinase enzyme and decomposes in the fungal cell wall. The presence of the chitins generally helps in inducing the activation of pathogenesis related (PR) protein. Activation of this PR protein helps in two ways. First, it catalyzes the hydrolysis process, and second, it activates the phenylalanine ammonia-lyase enzyme involved in triggering and controlling defense mechanisms through the phenylpropanoid pathway [64]. Chitosan concentrations can cause rapid potassium efflux, leading to a decrease in H+-ATPase that accumulates protons inside the cell, causing negative changes in H+/K+ exchange transport across the membrane of *Rhizopus stolonifera* [65]. Chitosan’s mechanism of action has been brilliantly explained by various researchers. With combined treatment (1% seed treatment + 0.5% foliar application) with chitosan, the investigator recorded the lowest DI (10.08%) at his 90 DAT (days after treatment) [38]. Chitosan application significantly reduced the release of zoospores from the sporangia of *Phytophthora capsici* and also restricted the movement of the zoospores and transformed them into round cystospores [43]. Chitosan affects germination and hyphal morphology of economically important post-harvest fungal pathogens (e.g., *Rhizopus stolonifer* and *Botrytis cinerea*) [66,67]. This polymer also hampers the growth of different plant pathogenic and mycoparasitic fungi including *Alternaria* spp., *Colletotrichium* spp., or *Trichoderma* spp.). Chitosan was recently found to permeabilize the plasma membrane of N.crass and flow cytometry was performed. This triggers intracellular production of reactive oxygen species (ROS) and cell death [68]. RNAseq data and gene ontology (GO) analysis revealed oxidoreductase activity, plasma membrane, and transport as the main categories induced by chitosan [69]. Chitosan also enhances oxidative metabolism, respiration, and GO transport functions in the plasma membrane of the model yeast Saccharomyces cerevisiae, and the stress response and cell wall integrity genes were also identified to be induced by chitosan [70].

b. Molecular mechanism of chitosan on sensitive and tolerant fungi

Chitosan generally exhibits different modes of action in host plants and fungi. In fungi, plasma membrane fluidity generally determines the susceptibility of fungi to chitosan. High plasma membrane fluidity is caused by the presence of polyunsaturated fatty acids (FFAs) such as linolenic acid and is observed in chitosan-sensitive fungi such as *Neurospora crassa* [71]. Lopez-Moya et al. (2019) described that the opposite mechanism of action can be observed in the plasma membrane of chitosan-resistant fungi with low fluidity and saturated FFAs [72]. This fluidity effect is the single key factor that triggers ROS production in fungal cells, leading to cell death [73]. In addition, genes encoding lipase class III, monosaccharide transport, and glutathione transferase play a role in targeting chitosan in fungi, ensuring plasma membrane repair and buffering of ROS, leading to chitosan damage to cells and modulates the antifungal activity of fungi [69]. In chitosan-resistant fungi, chitosan is degraded by the action of chitin deactylase or chitosanase [18]. Chitosan has been shown to be self-defeating against pathogens and, conversely, to support biocontrol mechanism fungi [74].

A summary of the chitosan mode of action can be visualized by Figure 4.

## 5. Conclusions

Since the discovery of chitosan, researchers have conducted extensive research to understand its new properties, new application methods and its efficacy. Study of chitosan has repeatedly demonstrated its versatile properties for combating plant diseases through its broad-spectrum anti phytopathogenic activity and by inducing the plant’s own defense activity. For this reason, in this era in chitosan is being highlighted as a legitimate alternative to chemicals and greatly ensuring food and environmental safety.

## 6. Future Aspect

Although extensive research has been conducted, further research is needed to identify the exact mechanism of action of chitosan in combating pathogens. Studies of the efficacy of chitosan to combat viral and prokaryotic plant pathogenic diseases need to be more comprehensive. The persistence of induced defense in the host, gene expression after chitosan application and inherited changes in defense are all awaiting exploitation and staining by researchers.

## Figures and Tables

**Figure 1 life-12-01908-f001:**
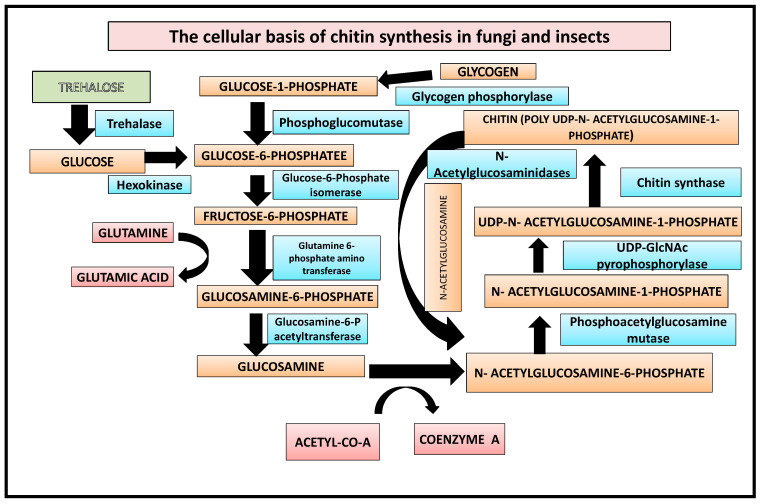
Biochemical basis of chitin synthesis in fungi and insects.

**Figure 2 life-12-01908-f002:**
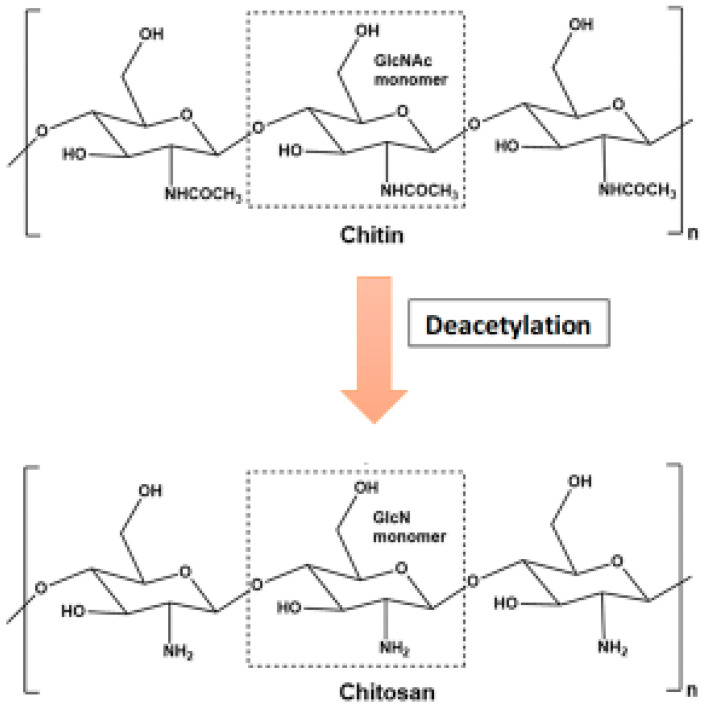
Conversion of chitosan from chitin through deacetylation.

**Figure 3 life-12-01908-f003:**
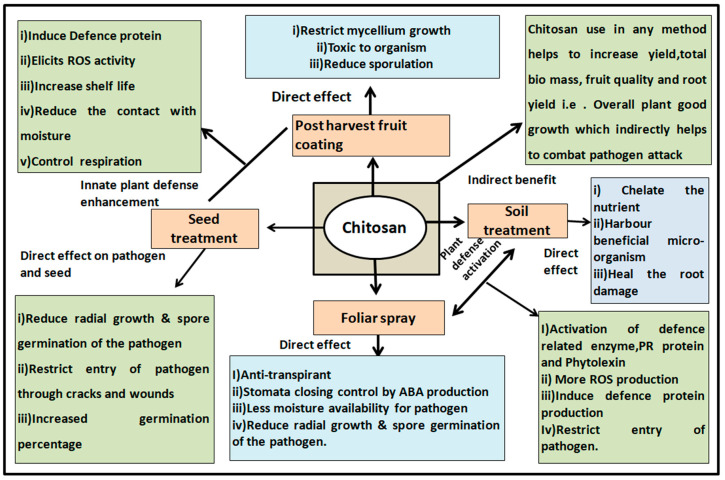
Methods of use of chitosan and their impact on host and pathogen both directly and indirectly in inducing defence mechanism or direct killing.

**Figure 4 life-12-01908-f004:**
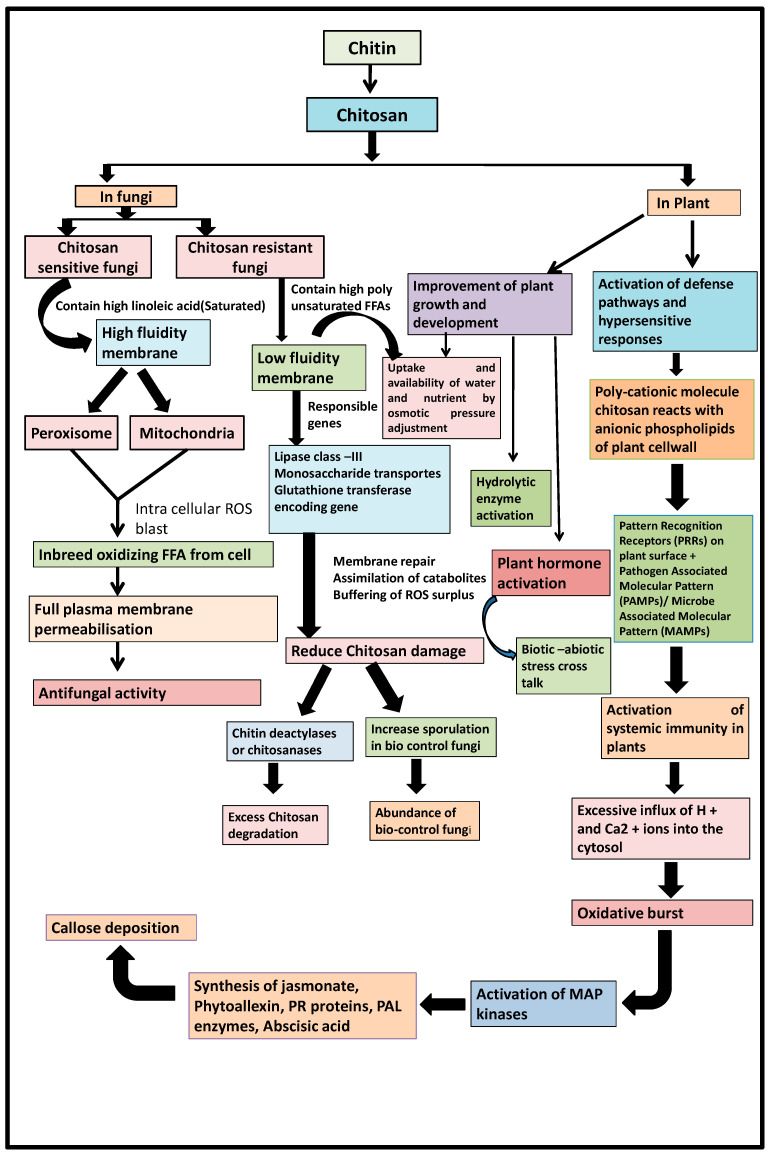
A summary of the chitosan mode of action.

**Table 1 life-12-01908-t001:** Efficacy of chitosan and chitin derivatives against different microorganisms.

Sl.No	Chitin Derivative	Target Pathogen	Remarks	References
1	Chitosan supplemented with 0.05% boron and 0.05%	*Pseudomonas* *syringae pv. actinidiae*	Inhibition of the growth of bacteriumin in vitro condition	[32]
2	Chitin (Shrimp shell)	*Aspergillus fumigatus*, *Aspergillus flavus* and *Aspergillus niger*	In in vitro condition, highest inhibition (19.5 mm) in case of *Aspergillus fumigatus*	[33]
3	Chitin methanol extract	*Aspergillus fumigatus*, *Aspergillus flavus* and *Aspergillus niger*	Highest inhibition (16.5 mm) in case of *A. fumigatus*	[33]
4	Chitosan	*Aspergillus fumigates*, *Aspergillus flavus* and *Aspergillus niger*	Highest inhibition (14 mm) in case of *A. fumigatus*	[33]
5	Chitosan	*Alternaria solani*	Complete inhibition in in vitro condition at 5.0 g/lit	[1]
6	Chitin (CT), 6-amino-chitin (NCT) and 3,6-diamino-chitin (DNCT)	*F. oxysporum f.* sp. *cucumerium*, *B. cinerea*, *C. lagenarium*, *P. asparagi*, *F. oxysporum f. niveum*, and *G. zeae*	In in vitro condition, DNCT showed Highest inhibition zone (11.4–20.4 mm) > NCT > CT	[34]
7	Chitosan	*Aspergillus flavus*, *Rhizoctonia solani* and *Alterneria alterneta*	Growth inhibition was highest in case of *Aspergillus flavus* (10.66 mm.)	[35]
8	Chitosan-polyacrylic acid nanoparticles	*Aspergillus flavus*, *Fusarium oxysporum*, *Fusarium solani*, *Aspergillus terreus*, *Alternaria tenuis*, *Aspergillus niger* and *Sclerotium rolfsii*	Inhibition percentage was highest in case of *Aspergillus flavus* (60%),	[36]
9	Chitosan	*F. proliferatum* and *F. verticillioides*	Reduce deoxynivalenol (DON) and fumonisin (FBs) production on irradiated maize and wheat grains and growth rates of both the pathogens decreased.	[37]
10	Chitosan	*Colletotrichum capsici*	7.67% post-emergence seedling mortality where Seeds were treated with 1% chitosan.	[38]
11	Chitosan	*Fusarium oxysporum radicisycopersici*, *F. oxysporum lycopersici*, *F. solani, Rhizoctonia solani*, *Sclerotium rolfsii*, *Macrophomina phaseolinae*, *Pythium* sp. and *Phytophthora* sp.	In 5 g/lit concentration, 100% inhibition can occur against every tested pathogen	[39]
12	Chitosan	*Fusarium oxysporum f.* sp. *radicislycopersici*	16.60% and 42.8% reduction of disease severity in application of chitosan 1 g/lit and *T. harzianum* + Chitosan 1.0 g/lit	[40]

**Table 2 life-12-01908-t002:** Chitosan seed treatment in different crops and their efficacy.

Sl. No	Crop Name	Respective Dose	Efficiency on Target Pathogen	Reference
1	Fenugreek	2.0 g/lit	In pot and field studies, seeds treated with chitosan greatly reduced root rot disease severity of *Fusarium solani*	[1]
2	Potato	4.0 g/lit of acetic acid-distilled water solution	Reduced dry rot severity observed in case of *F. oxysporum* (60.0%)and *F. sambucinum* (48.2%) by chitosan treatment.	[42]
3	Chilli	1%	100% mycelium growth inhibition and the lowest (7.67%) post-emergence seedling mortality was observed against *Colletotrichum capsici*	[38]
5	Cucumber	500 ppm	500 ppm chitosan seed treatment showed 100% disease resistance against damping off caused by *Phytophthora capsici*	[43]

**Table 3 life-12-01908-t003:** Foliar spray of chitosan and their efficacy.

Sl.No	Crop	Effective Dose	Pathogen	Activity	Reference
1	Tomato	0.5 g/lit	*Rhizoctonia solani*	58.8% disease reduction in Pre-emergence damping off after 10 days	[40]
2	Cucumber	0.05–0.1%	*Colletotrichum* sp.	Disease control and reduction of lesion than the untreated one	[49]
3	Tea	0.01%	*Exobasidium vexans*	67.73% less disease incidence than control	[50]
4	Turmeric	0.1%	*Pythium aphanidermatum*	Reduced disease severity and increased chitinase activity.	[51]
5	Grape	0.8%	*Plasmopara vitccola*	81% disease reduction	[52]

**Table 4 life-12-01908-t004:** Postharvest disease control by chitosan effect in different crops.

Sl.No	Crop	Effective Dose	Pathogen	Activity	Reference
1	Pomegranate	0.1–10 g/lit of chitosan	*Botrytis* sp., *Penicillium* sp. and *Pilidiella granati*	Reduced rot incidence by 18–66%	[53]
2	Jujube fruit.	20 mg/ml	*Penicillium expansum*	More than 80% inhibition of incidence one day after treatment.	[54]
3	Kiwi fruit	5 gm/lit	*Gray mold* *(B. cinerea)* and *blue mold (P. expansum)*	Disease Incidence was 46% (gray mold) and 65% (blue mold) comparing to untreated control.	[55]
4	Peach	Chitosan and oligochitosan 5 g/lit	*Brown rot (Monilinia fruiticola)*	Disease incidence drastically reduced and in both the cases only 20% DI occurred.	[56]
5	Rose Apple	2%	*Penecillium expansum*	Disease incidence was only 14% which was 24% less than control	[57]

## Data Availability

Not applicable.

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
