# Peer review of "Chitosan: An Autocidal Molecule of Plant Pathogenic Fungus"

_life, 2022, doi:10.3390/life12111908_

Round 1
Reviewer 1 Report
the manuscript sounds acceptable but the language editing needs to be revised
Author Response
Respected sir
We have changed our manuscript as per your suggestion and we also have taken the help of English Native person for the betterment of our manuscript.
Thank you so much for your valuable suggestion.
Reviewer 2 Report
The review article is aimed at describing the versatile employments of chitosan in plant defense. As such it is worthy of attention, but, to be accepted, it necessarily needs a thorough revision of grammar and punctuation, furthermore presenting some sentences that did not appear clear to me in the sense.
Please find the attached file for the grammar problems evidenced in orange. Please check all the singular/plural forms, the missing spaces and the misspelled words. Vulgar names should not be written in italics.
General comments:
Introduction:
-the first 5 lines are about climatic change, how does it ties to the following sentence regarding the discovery of chitosan? And what do you exactly mean with “a biologically, chemically, and physically diverse compound”.
-Lines 34-37. You used twice the expression “ on the other hand”, please change the second one with something like “consequently”.
- Please clarify the sentence “has a similar prevalence to cellulose”…did you mean “among the most abundant polysaccharides in nature after cellulose”?
-line 43 “Other chitin related compounds” you have not introduced this part (that chitosan is a chitinrelated compound).
-STRUCTURE AND FORMATION OF INSECT CHITIN: why did you insert this part? This review is not about chitin. Please remove all the unnecessary parts related to chitin, even in the subsequent paragraph “Deacetylation and Hydrolysis of Chitin/Chitosan”.
-Figure 2 presents words written in French, please modify.
-Tables 2,3 and 4. Please use univocal unit of measures; in the same table you reported “g/L”, “gL-1” and g/lit
-Line 189 “some beneficial micro organism like Bacillus” which Bacillus?
-line 192. J2, please specify the meaning.
-Lines 215-216. “As like as different method of chitosan use, the mode of action of chitosan regarding plant disease control has also multidisciplinary ways.” Multidisciplinary? Please reformulate.
-Line 234 “Chitosan treatment recorded 89% seed germination…” 89% compared to?
-Lines 260-265 “Chitin is a structural component of insect exoskeleton and fungal cell wall hydrolysed by chitinase that decompose the fungal cell wall. Chitin can induce some pathogenesis related protein that catalyse the hydrolysis process and on the other hand activate the enzyme phenylalanine ammonia-lyase that triggered and controlled defence mechanism through phenyl propanoid pathway. (Hadwiger 2013, Abbasi et al. 2009) .Changes in H+ -ATPase’s activity through plasma membrane is also an driven factor for plant pathogens survival.” This part is superfluous; please delete/reformulate in a more pertinent way.
-Line 269 “inner secret” what do you mean?
-Line 271 “0.5% foliar spray of chitosan showed lowest DI (10.08%) at 90 DAT “ Please rewrite and specify the meaning of DI and DAT
-Lines 289-292 “.Chitosan sensitive fungi like Neurospora crassa having high fluidity in plasma membrane due to the presence of polyunsaturated free fatty acid(FFA) like linolenic acid” Please reformulate
-Lines 292-293 “Lopez-Moya et al. (2019) also nicely described this molecular mechanism of chitosan interaction very nicely and…”. Please avoid both nicely and very nicely (especially in the same sentence)
Line 310 “newly evolved mode of action” what did you mean? Please clarify

Author Response
Respected sir
We have changed our manuscript as per your suggestion.

Reviewer 3 Report
Manuscript Title: Chitosan: An autocidal molecule of plant pathogenic fungus
life-1938579 contributes give the valuable information to the researchers and readers. The subject of the manuscript is consistent with the scope of the Journal. Thus, I suggested that the manuscript need to be major revised before it is accepted by this journal.
1. The manuscript (MS) let down by poor language. This drawback cause the perceived inaccurate and imperfect scientific style throughout all the MS sections preventing the readers to observe the MS novelty and challenging content. I could recommend to the editor that the authors have their paper language edited before resubmission, if applicable.
2. More literature information should be reviewed,such as Carbohyd. Polym, 2017, 164, 268–283; Agriculture, 2020, 10, 624; Not. Sci. Biol, 2021, 49, 1–15; Antibiotics, 2021, 10, 1449; Agriculture, 2022, 12, 373; Biomolecules, 2022, 12, 500; Int. J. Mol. Sci. 2019, 20, 332;
Author Response
Respected sir
We have changed our manuscript as per your suggestion and also have taken help from the English Native writer for the betterment of our manuscript. We added reference(Li et al.2021) from your suggesting reference list.
Thank you so much for your valuable suggestions.
Round 2
Reviewer 3 Report
I have no comments.